

Computer Science

# Landmark-free, parametric hypothesis tests regarding two-dimensional contour shapes using coherent point drift registration and statistical parametric mapping

Todd C. Pataky[1], Masahide Yagi[1], Noriaki Ichihashi[1] and Philip G. Cox[2,3]

[1] Department of Human Health Sciences, Kyoto University, Kyoto, Japan
[2] Department of Archaeology, University of York, York, United Kingdom
[3] Hull York Medical School, University of York, York, United Kingdom

## ABSTRACT

This paper proposes a computational framework for automated, landmark-free hypothesis testing of 2D contour shapes (i.e., shape outlines), and implements one realization of that framework. The proposed framework consists of point set registration, point correspondence determination, and parametric full-shape hypothesis testing. The results are calculated quickly (<2 s), yield morphologically rich detail in an easy-to-understand visualization, and are complimented by parametrically (or nonparametrically) calculated probability values. These probability values represent the likelihood that, in the absence of a true shape effect, smooth, random Gaussian shape changes would yield an effect as large as the observed one. This proposed framework nevertheless possesses a number of limitations, including sensitivity to algorithm parameters. As a number of algorithms and algorithm parameters could be substituted at each stage in the proposed data processing chain, sensitivity analysis would be necessary for robust statistical conclusions. In this paper, the proposed technique is applied to nine public datasets using a two-sample design, and an ANCOVA design is then applied to a synthetic dataset to demonstrate how the proposed method generalizes to the family of classical hypothesis tests. Extension to the analysis of 3D shapes is discussed.

Corresponding author
Todd C. Pataky,
pataky.todd.2m@kyoto-u.ac.jp

## INTRODUCTION

The statistical analysis of shape variation is relevant to a wide variety of academic fields including: evolutionary biology (*Mitteroecker & Gunz, 2009*), biomechanics (*Pedoia et al., 2017*), computer vision (*Murphy-Chutorian & Trivedi, 2008*), and many others (*Da Costa & Cesar, 2000*; *Rohlf & Marcus, 1993*; *Adams, Rohlf & Slice, 2004*; *Adams, Rohlf & Slice, 2013*). A key methodological framework for the statistical analysis of shape to have emerged in the literature is Geometric Morphometrics (*Corti, 1993*; *Bookstein, 1996*; *Slice,*

*2007*; *Zelditch, Swiderski & Sheets, 2012*). Geometric Morphometrics consists of a variety of statistical techniques, ranging from classical hypothesis testing (e.g., *Goodall, 1991*) and classical dimensionality reduction techniques like principal component analysis (*Adams, Rohlf & Slice, 2004*) to machine learning techniques like unsupervised clustering (*Renaud et al., 2005*). This paper is concerned primarily with classical hypothesis testing as it pertains to shape analysis.

A common geometric morphometric approach to classical hypothesis testing regarding group differences (depicted in Fig. 1A), consists of: (1) landmark definition, (2) spatial registration, and (3) Procrustes ANOVA (*Goodall, 1991*). Landmark definition refers to the manual identification and digitizing (i.e., XYZ coordinate specification) of homologous points on multiple objects, for example the corners on polyhedra. Spatial registration refers to the optimal, non-shearing affine alignment of a set of landmarks; that is, the optimal translation, rotation and scaling of each set of landmarks is calculated so that the landmarks are optimally aligned in space. Procrustes ANOVA is effectively equivalent to classical ANOVA, where Procrustes distance is the dependent variable (*Zelditch, Swiderski & Sheets, 2012*).

Landmarks with evolutionary, developmental or functional homology are essential for accurate interpretation of results (*Hallgrimsson et al., 2015*), especially for biological studies which seek to understand morphological variation in the context of evolution (e.g., *Stayton, 2005*; *Morgan, 2009*; *Casanovas-Vilar & Van Dam, 2013*; *Dumont et al., 2016*; *Page & Cooper, 2017*), ontogeny (e.g., *Klingenberg & McIntyre, 1998*; *Mitteroecker et al., 2004*; *Singleton, 2015*) or function (e.g., *Terhune, Cooke & Otárola-Castillo, 2015*; *Toro-Ibacache, Muñoz & O'Higgins, 2016*). A key practical advantage of landmark approaches is that they impose problem tractability; they convert abstract, usually high-dimensional shape representations including images, scans and line contours, to a relatively small set of numeric coordinates which can be assembled into readily processable data formats like text files and spreadsheets. This practical advantage is reinforced by well-established statistical theory (e.g., *Gower, 1975*; *Kendall, 1977*; *Kendall, 1984*; *Kendall, 1985*; *Kent, 1994*; *Rohlf, 1999*) which describes a comprehensive solution for dealing with shape data's inherent dimensionality problem (*Rohlf, 2000b*; *Rohlf, 2000a*; *Collyer, Sekora & Adams, 2015*).

A common approach to landmark-based hypothesis testing is Procrustes ANOVA. While landmark data themselves are multivariate (i.e., multiple landmarks, each with multiple coordinates are used to describe a single shape), Procrustes ANOVA uses a univariate metric (Procrustes distance) to test shape-relevant hypotheses. One problem with this approach is that a single value is likely inadequate to fully characterize shape effects. Many other shape descriptors exist (*Kurnianggoro, Wahyono & Jo, 2018*), including both univariate metrics like eccentricity and multivariate metrics like geometric moments (*Zhang & Lu, 2004*). It has been argued that focus on relatively low dimensional shape metrics like these is necessary in order to achieve suitable statistical power, with the assumption that too many variables relative to the number of phenotypes can preclude hypothesis testing via parametric methods, especially for small samples (*Collyer, Sekora & Adams, 2015*); one aim of this paper is to challenge that assertion, and to show that hypothesis testing is

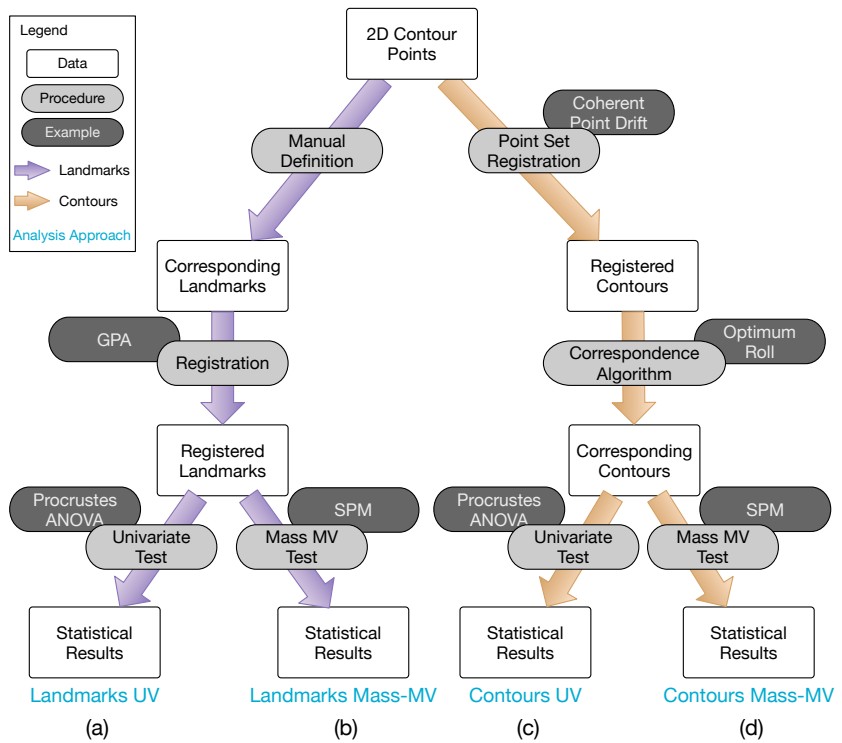

**Figure 1** **Overview of 2D contour data processing approaches employed in this paper.** (A) The most common analysis approach, consisting of Generalized Procrustes Analysis (GPA) and Procrustes ANOVA for landmarks. (B) Same as (A), but using mass-multivariate (MV) analysis instead of Procrustes ANOVA's univariate (UV) approach. (C) and (D) are conceptually equivalent to (A) and (B), respectively, but operate on full contour data instead of landmark data, and can also be fully algorithmic. Statistical Parametric Mapping (SPM) is a methodology for mass-MV analysis of continuous data. See text for more details.

indeed possible for even high-dimensional representations of shape, and with suitably high statistical power for even relatively small sample sizes.

A related sample size-relevant theoretical limitation of Procrustes ANOVA is that there is no known parametric solution to the underlying Procrustes distance probability distributions. Consequently, statistical inference is conducted nonparametrically, often using bootstrapping or permutation techniques (*Zelditch, Swiderski & Sheets, 2012* pp. 248–259). These nonparametric procedures are inherently poor for small sample sizes (*Anderson & Braak, 2003*; *Brombin & Salmaso, 2009*) because the probability distributions are constructed empirically and numerically, using the actual data, and both the precision and accuracy of these nonparametrically constructed distributions can decrease substantially with small sample sizes.

A variety of landmark-free or landmark-minimal methods also exist, including for example techniques that fit mathematical curves to shape outlines (*Rohlf, 1990*). One technique that has been particularly widely used is elliptical Fourier analysis (*Claude, 2013*; *Bonhomme et al., 2014*), which considers the spatial relations amongst neighboring points, and characterizes the spatial frequencies along the contour perimeter as a change-relevant

representation of shape. Elliptical Fourier analysis has been frequently employed to analyse structures on which few homologous landmarks can be identified such as fins, jaws and teeth (e.g., *Fu et al., 2016*; *Hill et al., 2018*; *Cullen & Marshall, 2019*). These methods are highly relevant to the methods described in this paper, in that they deal with original, high-dimensional shape data like 2D contours and 3D surface scans.

While landmark-free or landmark-minimal methods initially operate on original high-dimensional shape data, they tend to use much lower-dimensional representations of shape when conducting classical hypothesis testing. For example, elliptical Fourier analysis tends to conduct hypothesis testing using a relatively small number (fewer than ten) harmonic coefficients (*Bonhomme et al., 2014*). Common landmark and landmark-free methods are thus similar from from a hypothesis testing perspective in that the hypothesis tests ultimately pertain to relatively low-dimensional shape metrics.

The main aim of this paper was to show that classical hypothesis testing is possible on original, high-dimensional shape data, and in particular on continuous surfaces, without the need for low-dimensional shape representations, and with suitably high power even for analyses of relatively small samples. The methodology, which we refer to as 'continuous, mass-multivariate analysis' consists of a number of previously described techniques including: (1) point set registration, (2) correspondence, and (3) mass-multivariate hypothesis testing. This combination of techniques allows one to conduct landmark-free hypothesis testing on original surface shapes. For interpretive convenience we limit focus to 2D contours (*Bookstein, 1997*; *Carlier et al., 2016*), but in the Discussion describe how the proposed methodology can be applied to 3D surfaces.

## METHODS

Analyses were conducted in Python 3.6.10 (*Van Rossum, 2019*) using Anaconda 3.6.10 (*Anaconda, 2020*) and in R 3.6.2 (*R Core Team, 2019*) . Data processing scripts are available along with all original and processed data in this project's public repository at: https://github.com/0todd0000/lmfree2d.

### Datasets

Nine datasets were analyzed (Fig. 2). All datasets were taken from the open-source 2D Shape Structure database (*Carlier et al., 2016*) (http://2dshapesstructure.github.io). The database consists of 70 different shape classes. Inclusion criteria for shape class were: (i) qualitatively similar geometry in at least 10 shapes (Fig. 3), and (ii) at least four readily identifiable landmarks for all contour shapes.

Each dataset consisted of 20 contour shapes, where a 'dataset' represents a shape class (e.g., 'Bell' or 'Face') and individual shapes represent morphological variation within that shape class. We manually selected ten shapes from each dataset in a pseudo-random manner in order to span a range of effect sizes; in the Results, note that $p$ values span a wide range ($p < 0.001$ to $p > 0.9$). We selected just ten shapes primarily because it has been suggested that parametric procedures are unsuitable for the morphological analyses of small samples (*Collyer, Sekora & Adams, 2015*), and we wished to demonstrate that the proposed parametric technique is indeed sufficiently powerful for small-sample analyses.

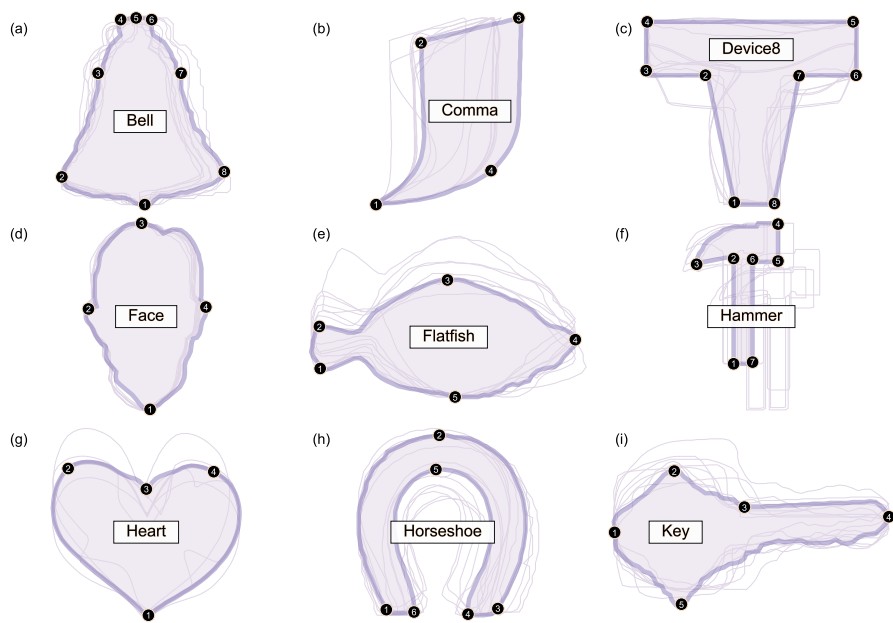

**Figure 2** **Overview of analyzed datasets.** All contour data are available in the 2D Shape Structure Dataset (*Carlier et al., 2016*). (A–I) For each dataset in this figure, one representative shape is highlighted, along with its numbered landmarks. Note that shape variance ranges from relatively small (e.g., Bell, Face) to relatively large (e.g., Device8, Heart).

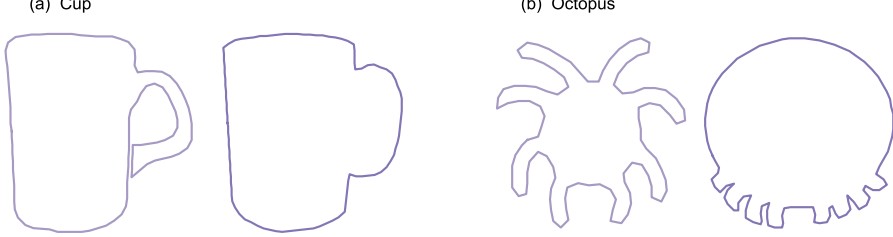

**Figure 3** **Shape class exclusion examples.** Shape classes were excluded if they contained shapes with qualitatively different contour geometry. For example: (A) the 'cup' class was excluded because some shapes had unattached handles with holes and others had attached handles without holes. (B) The 'octopus' class was excluded because the eight appendages appeared in non-homologous locations.

Secondary reasons for considering just 10 shapes included: (1) qualitatively different within-class geometry, implying that statistical comparisons would be dubious if all 20 shapes were used, (2) inconsistent curvature characteristics (e.g., some with sharp corners, others with no discernible corners), implying landmarking difficulties, and (3) untrue contour data (e.g., internal loops and thus non-convex polygons) implying that contour parameterization was not possible for all shapes.

Two-sample tests were conducted on each dataset using the four approaches as described below. For replicability, the final set of ten shapes selected for analysis from each class are redistributed in this project's repository at: https://github.com/0todd0000/lmfree2d. Note

**Table 1  Dataset count summary.** Point counts refer to the original data from *Carlier et al. (2016)*.

| Name | Shapes | Points | | | Landmarks |
|---|---|---|---|---|---|
| | | **Min** | **Median** | **Max** | |
| Bell | 10 | 101 | 104 | 185 | 8 |
| Comma | 10 | 101 | 104 | 108 | 4 |
| Device8 | 10 | 101 | 104 | 107 | 8 |
| Face | 10 | 103 | 104 | 106 | 4 |
| Flatfish | 10 | 100 | 102 | 112 | 5 |
| Hammer | 10 | 102 | 105 | 119 | 7 |
| Heart | 10 | 102 | 105 | 109 | 4 |
| Horseshoe | 10 | 106 | 109 | 128 | 6 |
| Key | 10 | 103 | 106 | 115 | 5 |

that the ultimately selected contours had a variable number of contour points within each dataset (Table 1).

## Data processing

The 2D contour shape data were analyzed using four related approaches, consisting of the four combinations of (i) landmarks vs. contours, and (ii) univariate (UV) vs. mass-multivariate (mass-MV). These four approaches are summarized in Fig. 1. The Landmarks-UV approach (Fig. 1A) is common in the literature, none of the other approaches is common. The primary purpose of this study was to compare and contrast the Landmarks-UV and Contours-MassMV approaches (Figs. 1A, 1D). We also employed intermediary approaches (Figs. 1B, 1C) to more clearly highlight the differences between the two main approaches.

### Landmarks univariate (UV) analysis

Landmarks were defined for each dataset as depicted in Fig. 2. Both the number of landmarks (Table 1) and their locations were selected in an *ad hoc* manner, with the qualitative requirement of readily identifiable, homologous locations. The ultimately selected landmarks arguably span a representative range of landmarking possibilities.

One operator used a mouse to manually digitize the landmarks for each of the 90 shapes (10 shapes for each of 9 datasets). The operator was ignorant of the final shape groupings for the ultimate two-sample tests (see below), implying that the landmarking was performed without grouping bias.

The landmarks were spatially registered using Generalized Procrustes Analysis (GPA) (*Gower, 1975*), and the resulting registered landmarks were analyzed in a univariate manner, using Procrustes ANOVA (*Goodall, 1991*)—a method which considers the variance in the Procrustes distance across a dataset. Note that the Procrustes distance is a scalar quantity that summarizes shape difference, and thus that this method is univariate. GPA and Procrustes ANOVA were both conducted using the **geomorph** package for R (*Adams & Otárola-Castillo, 2013*).

### Landmarks mass-multivariate (mass-MV) analysis

This approach was identical to the Landmarks-UV approach described above, except for statistical analysis. The two-sample Hotelling's $T^2$ statistic was calculated for each landmark according to its definition:

$$T_i^2 = \frac{n_1 n_2}{n_1 + n_2} \left( \bar{r}_{1i} - \bar{r}_{2i} \right)^\top W_i^{-1} \left( \bar{r}_{1i} - \bar{r}_{2i} \right) \tag{1}$$

where $i$ indexes landmarks, the subscripts "1" and "2" index the two groups, $n$ is sample size, $\bar{r}_i$ is the mean position vector of landmark $i$, and $W_i$ is the pooled covariance matrix for landmark $i$:

$$W_i = \frac{1}{n_1 + n_2 - 2} \left( \sum_{j=1}^{n_1} (r_{1ij} - \bar{r}_{1i})(r_{1ij} - \bar{r}_{1i})^\top + \sum_{j=1}^{n_2} (r_{2ij} - \bar{r}_{2i})(r_{2ij} - \bar{r}_{2i})^\top \right) \tag{2}$$

where the $i$ index is dropped for convenience in Eq. (2).

Statistical inference was conducted in a mass-multivariate manner, using Statistical Parametric Mapping (SPM) (*Friston et al., 2007*). SPM bases statistical inferences on the distribution of the maximum $T^2$ value $(T_{\max}^2)$, which can be roughly interpreted as the largest landmark effect, and which is defined as:

$$T_{\max}^2 \equiv \max_{i \in L} T_i^2 \tag{3}$$

where $L$ is the number of landmarks.

SPM provides a parametric solution to the distribution of $T_{\max}^2$ under the null hypothesis, so significance can be assessed by determining where in this distribution the observed $T_{\max}^2$ lies. Classical hypothesis testing involves the calculation of a critical threshold $(T^2)_{\text{critical}}$, defined as the $(1-\alpha)$th percentile of this distribution, and all landmarks whose $T^2$ values exceed $(T^2)_{\text{critical}}$ are deemed significant at a Type I error rate of $\alpha$. This is a correction for multiple comparisons (i.e., across multiple landmarks) that is 'mass-multivariate' in the following sense: 'mass' refers to a family of tests, in this case a family of landmarks, and 'multivariate' refers to a multivariate dependent variable, in this case is a two-component position vector. This is similar to traditional corrections for multiple comparisons like Bonferroni corrections, with one key exception: rather than using the total number of landmarks $L$ as the basis for the multiple comparisons correction, as the Bonferroni correction does, SPM instead solves the mass-MV problem by assessing the correlation amongst neighboring landmarks or semilandmarks, and using the estimated correlation to provide a less severe correction than the Bonferroni correction, unless there is no correlation, in which case the SPM and Bonferroni corrections are equivalent.

### Contours univariate (UV) analysis

Similar to the Landmarks UV approach, this approach ultimately conducted Procrustes ANOVA, but did so on contour data rather than landmark data. This was achieved through two main processing steps: coherent point drift (CPD) point set registration (Fig. 4) and optimum roll correspondence (Fig. 5). Coherent point drift (CPD) (*Myronenko & Song, 2010*) is a point set registration algorithm that spatially aligns to sets of points that belong to

(a) Original                                    (b) CPD-registered

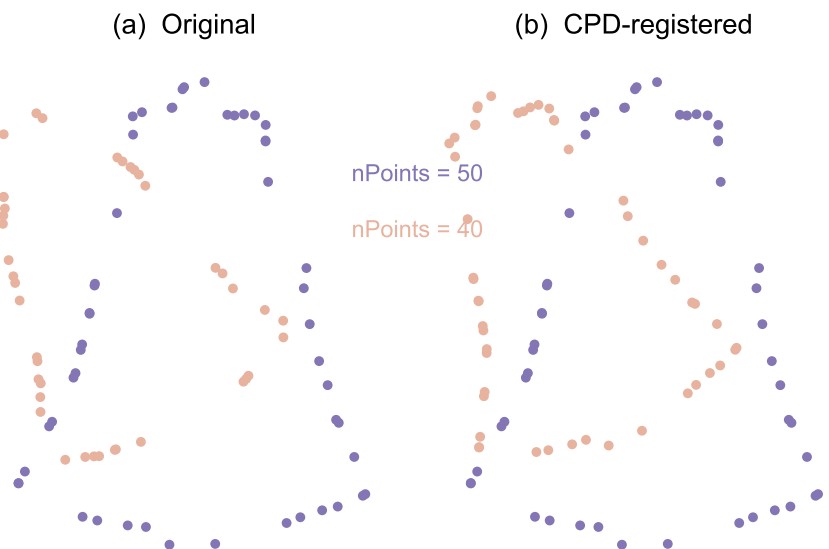

**Figure 4  Example point set registration using the coherent point drift (CPD) algorithm.** (A) Original. (B) CPD-registered. Note that CPD requires neither corresponding points, nor an equal number of points.

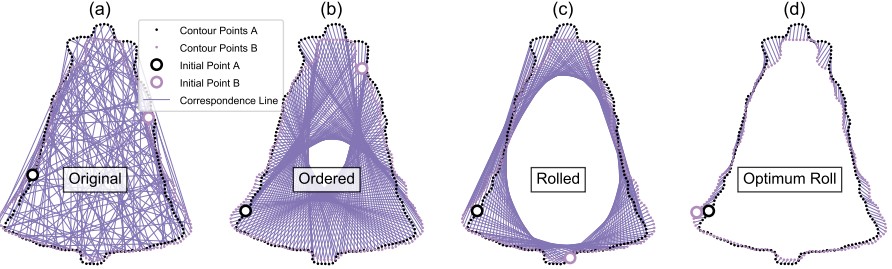

**Figure 5  Example optimum roll correspondence.** (A) Original data, consisting of an equal number of contour points, arranged in a random order. (B) Ordered points; clockwise along the contour. (C) Rolled points; moving the initial point of contour B brings the shapes into better correspondence. (D) Optimally rolled points; the total deformation energy across all points (i.e., the sum-of-squared correspondence line lengths) is minimum.

the same or a similar object. Neither an equal number of points nor homologous points are required (Fig. 4), making this approach useful for contours that have an arbitrary number of points.

Since contour points from arbitrary datasets may generally be unordered (Fig. 5A), we started our analyses by randomly ordering all contour points, then applying CPD to the unordered points. We acknowledge that many 2D contour datasets consist of ordered points—including those in the database used for this study (*Carlier et al., 2016*)—but since 3D surface points are much more likely to be unordered, we regard unordered point support as necessary for showing that the proposed method is generalizable to 3D analyses. Following CPD, we re-ordered the points using parametric surface modeling (*Bingol & Krishnamurthy, 2019*), which fits a curved line to the contour, and parameterizes

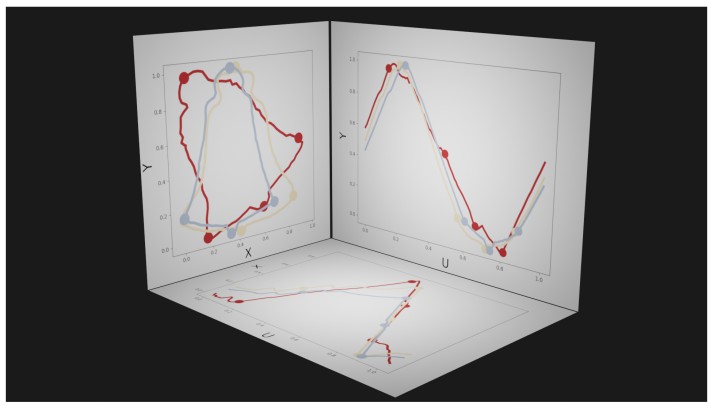

**Figure 6 Example parametric representations of 2D contour shape.** Dots represent manually defined landmarks, and are shown as visual references. Left panel (XY plane): the spatial plane in which shape data are conventionally presented. The three colors represent different shapes. Bottom panel (UX plane) and right panel (UY plane): abstract planes in which U represents the parametric position (from 0 to 1) along the contour; positions $U = 0$ and $U = 1$ are equivalent.

the contour using position $u$, where $u$ ranges from zero to one (Fig. 6). This contour parameterization results in a continuous representation of the contour, from which an arbitrary number of ordered points (Fig. 5B) can be used to discretize the contour of each shape for subsequent analysis. We used NURBS parameterization with B-spline interpolation (*Bingol & Krishnamurthy, 2019*) to calculate specific contour point locations. We then applied an optimum roll transformation, which found the value of $u$ for one contour that minimized the deformation energy across the two contours (Figs. 5C, 5D).

We repeated contour parameterization, ordering, and optimum roll correspondence across all contour shapes, using the shape with the maximum number of contour points in each dataset as the template shape to which the nine other shapes were registered. Note that this registration procedure is unrelated to the traditional landmark analyses described in 'Landmark UV analysis' above, for which an equal number of points is a requirement of registration and analysis. The correspondence analysis step resulted in an equal number of contour points, upon which we conducted Procrustes ANOVA.

### Contours mass-multivariate (mass-MV) analysis

This approach was identical to the Contours-UV approach, with the exception of statistical analysis, which we conducted using SPM as outlined above. Unlike the landmark data above, which are generally spatially disparate, contour points are spatially proximal, and neighboring points tend to displace in a correlated manner. For example, if one contour point in a specific shape lies above the mean point location, its immediate neighbors also tend to lie above the mean location). SPM leverages this correlation to reduce the severity of the multiple comparisons correction, and SPM solutions converge to a common $(T^2)_{\text{critical}}$ regardless of the number of contour points, provided the number of contour points is sufficiently large to embody the spatial frequencies of empirical interest, as outlined in classical signal processing theory (*Nyquist, 1928*).

**Table 2  Statistical results summary, probability values.** As nonparametric inference yielded similar p values (see Results), only parametric p values are reported in this table for brevity.

| Name | Landmarks | | Contours | |
|---|---|---|---|---|
| | UV | Mass-MV | UV | Mass-MV |
| Bell | 0.130 | 0.302 | 0.084 | **0.041** |
| Comma | 0.155 | 0.294 | 0.719 | 0.327 |
| Device8 | **0.022** | 0.214 | 0.433 | 0.681 |
| Face | **0.025** | 0.103 | 0.052 | **0.013** |
| Flatfish | **0.023** | **0.016** | **0.026** | **0.001** |
| Hammer | 0.708 | 0.206 | 0.417 | **<0.001** |
| Heart | 0.940 | 0.976 | 0.544 | **0.016** |
| Horseshoe | 0.084 | **0.008** | **0.006** | **0.001** |
| Key | 0.532 | 0.270 | **0.013** | **0.022** |

As SPM uses parametric inference to calculate the critical $T^2$ threshold, and Procrustes ANOVA uses nonparametric inference, we also conduct Contours Mass-MV analysis using statistical non-parametric mapping (*Nichols & Holmes, 2002*), which uses permutation to numerically build the $T^2_{\mathrm{max}}$ distribution under the null hypothesis. This permutation approach converges to the parametric solution when the residuals are normally distributed (i.e., point location variance follows an approximately bivariate Gaussian distribution). All SPM analyses were conducted in **spm1d** (*Pataky, 2012*); note that one-dimensional SPM is sufficient because the contour domain ($U$) is one-dimensional (Fig. 6).

## RESULTS

The four analyses approaches produced a range of $p$ values from very low ($p < 0.001$) to very high ($p > 0.9$), and even yielded a large range of $p$ values for single datasets (e.g., Heart: $0.016 < p < 0.940$) (Table 2). Of the nine datasets, only two yielded consistent hypothesis testing conclusions (at $\alpha = 0.05$) across the four analysis approaches: for the Comma dataset all approaches failed to reject the null hypothesis, and for the Flatfish dataset all approaches rejected the null hypothesis. The seven other datasets showed a range of disagreement amongst the methods. For example, for the Key dataset neither Landmarks approach reached significance, but both Contours approaches did reach significance. For the Hammer dataset, three approaches failed to reach significance, but the Contours Mass-MV approach produced a very low $p$ value ($p < 0.001$). The Landmarks approaches executed comparatively rapidly (~50 ms) compared to the Contours approaches (~2 s) (Table 3).

Since Procrustes ANOVA results are commonly used in the literature, and are summarized for the current study in Table 2, the remainder of the results considers the Mass-MV approaches' results. First, the Landmarks Mass-MV approach indicate a wide range of $T^2$ statistic values at each landmark (Fig. 7). For example, Landmark 5 in the Horseshoe dataset (Fig. 2) had a very high $T^2$ value, and all other landmarks had comparatively low $p$ values (Fig. 7). This suggests that (a) shape differences can be highly localized, and that (b) univariate methods that employ an overall shape change metric, like

**Table 3 Execution durations (unit: ms).** Averages across the nine datasets. Procrustes ANOVA (Proc-ANOVA) involved 1000 iterations for each dataset. Average SnPM durations (not shown in this table) were 344.0 and 6336.0 ms for Landmarks Mass-MV and Contours Mass-MV, respectively.

| Category | Procedure | Landmarks | | Contours | |
|---|---|---|---|---|---|
| | | UV | Mass-MV | UV | Mass-MV |
| *Registration | CPD | – | – | 414.1 | 414.1 |
| | Point Ordering | – | – | 327.9 | 327.9 |
| | Interpolation | – | – | 835.1 | 835.1 |
| | Correspondence | – | – | 40.9 | 40.9 |
| | GPA | 6.7 | 6.7 | 8.5 | – |
| Hypothesis test | Proc-ANOVA | 60.0 | – | 99.0 | – |
| | SPM | – | 39.3 | – | 66.8 |
| **Total** | | 66.7 | 46.0 | 1,725.5 | 1,684.8 |

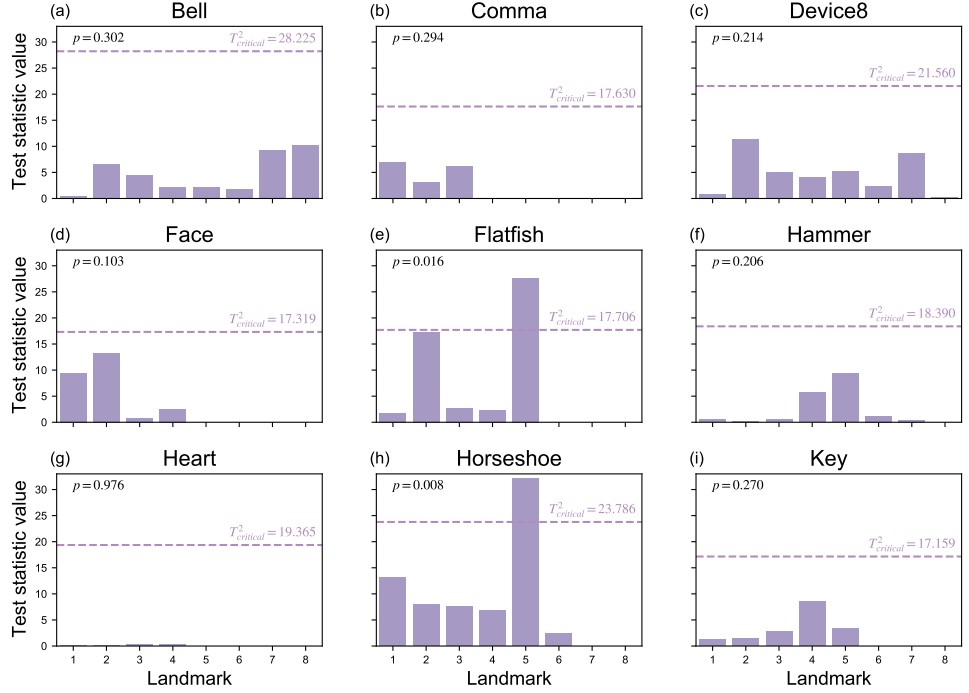

**Figure 7 Landmark results from mass-multivariate testing.** (A–I) Landmark-specific $T^2$ values are presented along with the critical threshold at $\alpha = 0.05$, and probability values for the overall mass-multivariate test.

Procrustes ANOVA, may not be able to detect these changes, even when the landmarks are identical (Table 2).

The Contour Mass-MV results showed little qualitative difference between parametric and nonparametric inference (Fig. 8), with minor exceptions regarding specific locations and spatial extent of supra-threshold contour points (e.g., Key, Horseshoe). Since this Contour Mass-MV approach is sensitive to point-specific variation, it was generally more

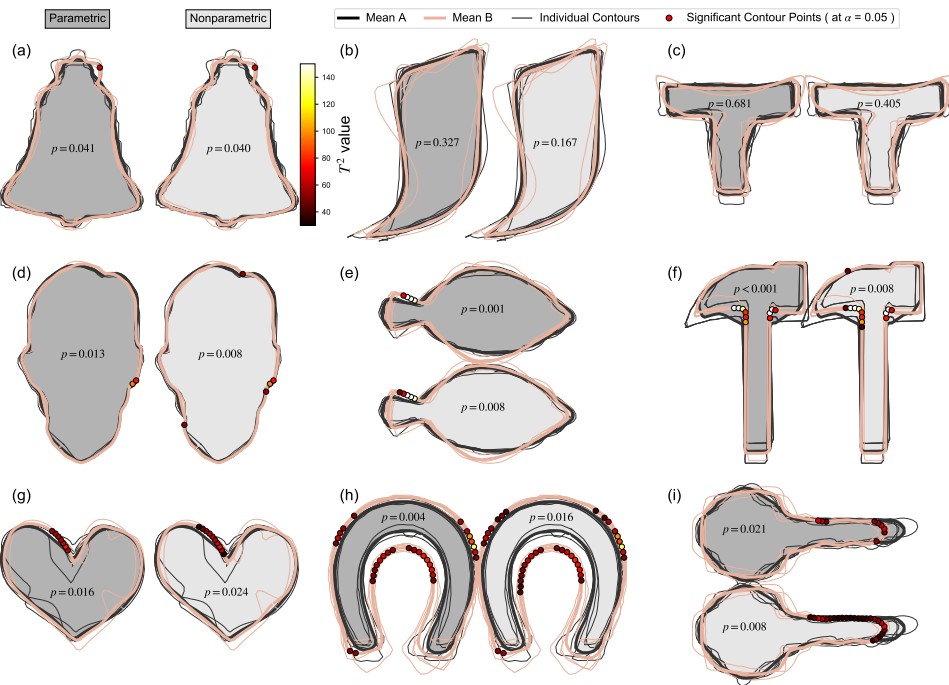

**Figure 8** **Contours mass-multivariate results using Statistical Parametric Mapping (SPM).** (A–I) Results for both parametric and nonparametric inference are shown. *P* values represent the probability that random variation in the Mean A contour would produce a deformation as large as in the observed Mean B, given the estimated contour variance. Dots on the Mean B contour represent contour points whose $T^2$ values exceeded the threshold for significance at $\alpha = 0.05$; if the maximum $T^2$ value did not reach this threshold, the *p* value is greater than $\alpha$, and no dots are shown.

sensitive at detecting changes, as shown in the relatively high rate of null hypothesis rejection relative to the other approaches (Table 2); that is, even though the Contours-UV and Contours Mass-MV approaches consider the same data, the latter reached significance more often than the former, implying that it is more sensitive to location-specific effects. Whether this sensitivity is a benefit or not is considered in the Discussion.

# DISCUSSION

## Main findings

This study's main result is the demonstration that it is possible to conduct fully automated, landmark-free, parametric hypothesis testing regarding whole 2D contour shapes, irrespective of the number of points and point ordering in the original contour data. These analyses can be executed relatively quickly; the current non-optimized implementation required less than 2 s for all analysis steps (Table 3). The proposed analysis framework (Fig. 1D) consists of families of previous techniques including: point set registration (e.g., *Myronenko & Song, 2010*), point correspondence algorithms (e.g., *Loy et al., 2000*; *Myronenko & Song, 2010*), and mass-multivariate testing (*Friston et al., 2007*; *Taylor & Worsley, 2008*; *Chung et al., 2010*), and some of these techniques have been used for classical hypothesis testing regarding shapes in the past (*Taylor & Worsley, 2008*; *Chung*

*et al., 2010*). A variety of landmark-free techniques have also been previously proposed (e.g., *Wuhrer, Shu & Xi, 2011*; *Taylor & Worsley, 2008*; *Chung et al., 2010*) Nevertheless, these techniques have not, to our knowledge, been previously combined into a general hypothesis testing framework—from raw data to statistical results—as depicted in Fig. 1D. The main novelty of this paper is thus the demonstration that it is possible to fully automate data processing from raw 2D contour data to final hypothesis testing results.

The second main novelty of this paper is the demonstration that parametric hypothesis testing is possible when conducted at the whole-contour level. We stress that 'possible' implies neither 'valid' nor 'appropriate'; demonstrating the validity and appropriateness of the proposed method would require substantial empirical efforts over a range of datasets, data modalities, experimental designs, and applications, in addition likely to simulation studies, and as such assessing validity and appropriateness are beyond the scope of this paper. We also stress that 'possible' does not imply that one should use the proposed technique in isolation. We believe that the proposed technique offers unique information that is complimentary to other techniques, and that ideally the results of multiple analysis techniques should be corroborated to build interpretive robustness.

The proposed analysis framework (Fig. 1D) offers various improvements over landmark analysis (Fig. 1A) including: (1) the modeling flexibility of classical hypothesis testing, (2) increased objectivity due to avoidance of subjective landmark definition and selection, (3) increased speed due to avoidance of manual work, and (4) unique, implicit morphological meaning in hypothesis testing results. We acknowledge that each of these improvements also involve limitations, and we address these limitations below. We stress that 'objectivity' implies none of 'accurate', 'useful' or 'interpretable'. We use 'objective' instead primarily to mean 'algorithmic'.

## Statistical Parametric Mapping (SPM)

SPM, like most parametric tests, assumes normality, so in this case SPM assumes that the spatial variability of all contour points are distributed in a bivariate Gaussian manner. This distributional assumption could be directly tested using distributional tests in a point-by-point manner. In this paper, instead of directly testing for distributional adherence, we instead tested the assumption indirectly, by conducting nonparametric tests (Fig. 8), which do not assume bivariate normality. In this case there were minor quantitative differences between the parametric and nonparametric results, but overall the qualitative interpretations were largely unaffected by the use of parametric vs. nonparametric analysis. This represents relatively strong (albeit indirect) evidence that the parametric approach's distributional assumptions are appropriate at best, or largely inconsequential at worst, for these particular datasets. This however does not imply that parametric inference is appropriate for all datasets, so distributional assumptions should generally be tested for all datasets, possibly indirectly through nonparametric tests like those conducted in this paper.

Although this paper considered only two-sample tests, SPM supports all classical hypothesis testing procedures, ranging from simple linear regression to MANCOVA (*Friston et al., 2007*), thereby making the proposed framework highly flexible to arbitrary

experimental designs. To emphasize this point, and how it may be valuable for general shape analysis, we conducted a set of supplementary analyses using synthetic data involving simple, circular shapes with controlled morphological effects (Figs. 9A, 9B). The controlled effects included a size-dependent signal, which was modeled using a Gaussian contour pulse that increased in amplitude with increasing shape size (as defined by the shape's average radius) (Fig. 9A), and a group-dependent signal, which was modeled similarly, but which was applied to just one of two hypothetical groups (Fig. 9B). To isolate and emphasize design flexibility, and to eliminate registration and correspondence as potential sources of error, we controlled both by sampling at 101 evenly distributed angular displacements with respect to the horizontal axis. We considered two MANCOVA possibilities: analysis of the original, unscaled dataset (Fig. 9A), and analysis of the scaled/registered dataset (Fig. 9B). We applied a single MANCOVA model, which modeled both shape size (i.e., mean shape radius) and group, and which thereby afforded consideration of both (1) size effects, with group effects linearly removed, and (2) group effects, with size effects linearly removed. Size effects for the original, unscaled data naturally showed very large test statistic values at all contour points (Fig. 9C). In contrast, size effects for the registered data correctly isolated the modeled size-dependent signal (Fig. 9D). Group effects were practically identical for both the original, unscaled data and the registered data (Figs. 9E, 9F), emphasizing the point that MANCOVA can be used to remove size-related effects in lieu of registration. More generally, this analysis shows that the proposed framework is highly flexible, and can be used with arbitrary continuous and categorical independent variables, provided these variables adhere to the requirements of classical linear design modeling. We nevertheless caution readers that the (Fig. 9) analyses consider close-to-ideal data, for which registration and correspondence are near-perfectly controlled. For real dataset analysis, both registration and correspondence generally introduce errors that may or not affect the ultimate hypothesis testing results. Results' sensitivity to data processing algorithms and their parameters must be considered in general analyses.

## Comparison with landmarking and other methods

The proposed methodology partially overcomes limitations of landmark selection, and the corresponding susceptibility to bias (*Arnqvist & Martensson, 1998*; *Rohlf, 2003*; *Fruciano, 2016*); shape-to-shape landmark identification is often manual and therefore subjective. Algorithmic landmark identification is nevertheless possible (*Claes et al., 2011*; *Strait & Kurtek, 2016*), and indeed modern machine learning techniques have been shown to substantially improve landmark detection, with the promise of eliminating landmark-associated subjectivity (*Morris, 2003*; *Young & Maga, 2015*; *Strait & Kurtek, 2016*; *Devine et al., 2020*). Like automated landmarking, the proposed method can be used with little-to-no subjective intervention, implying generally more repeatable results. Here 'objective' does not necessarily mean 'accurate' or 'appropriate'; it simply means that results are expected to be more reproducible than the results from more subjective methods. Determining the accuracy and appropriateness of all methods, including the proposed one, requires substantial empirical effort across a range of data modalities and applications.

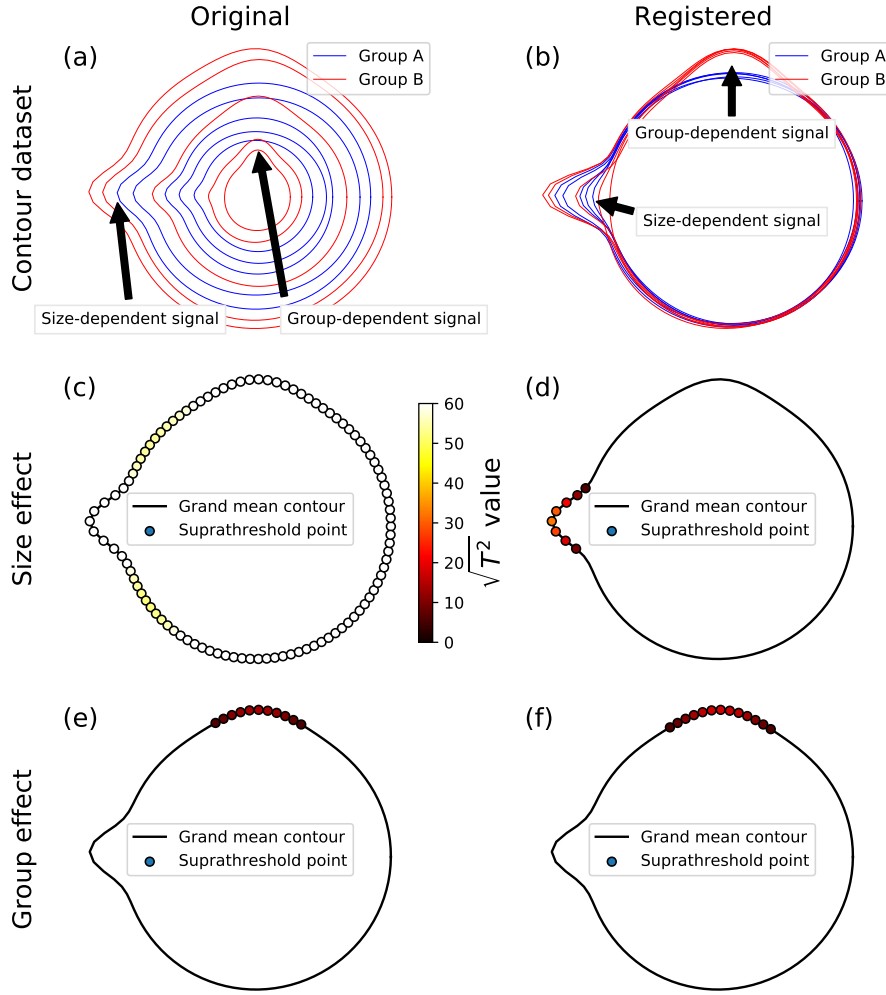

**Figure 9** **Example MANCOVA using synthetic data; for simplicity, data were generated to have (i) a relatively large signal:noise ratio, and (ii) close-to-perfect correspondence, by sampling at 101 equally spaced angular distances around the contour.** (A) The original contour dataset, consisting of five noisy circles for each of two groups, with systematically different mean radii, and also with both group- and size-dependent signal, where 'size' was considered to be the mean radius, and where 'signal' implies true morphological difference. Note that the size-dependent signal is more easily perceived in (A), and that the group-dependent signal is more easily perceived in the next panel. (B) Registered contours. (C, D) Size effects from MANCOVA for the original and registered data; the test statistic is presented as $\sqrt{T^2}$ because a linear $T^2$ scale would result in imperceivable color differences (i.e., the (C) points would be all white, and the points in the other panels would all be close-to-black). (E, F) Group effects from MANCOVA for the original and registered data; note that the (E) and (F) results are similar because MANCOVA accounts for size-related effects in the 'Original' data.

We also note that the proposed landmark-free approach is just one end of the spectrum, where manual landmark definition is the other, and that a variety of alternative techniques occupy positions between these two extremes. For example, semilandmarks (*Mitteroecker & Gunz, 2009*) provide an objective way to fill spatial gaps between landmarks, thereby creating a continuous surface. From the perspective of the proposed method,

semilandmarks represent the results of piecewise registration over the domain $u$ (Fig. 6), or equivalently a hybrid registration method consisting of both algorithmic and manual components (*Ramsay & Li, 1998*). As there are a plethora of automated techniques for geometrical matching (*Holden, 2008*), the proposed framework regards these techniques each as objective, substitutable, yet each imperfect components, whose assumptions and parameters could ultimately affect the final results. From this perspective, a second layer of objectivity could be added to the proposed framework, whereby different techniques and/or parameters are iteratively substituted in a sensitivity framework, to objectively discern the numerical stability of the final results, as well as the boundaries of that stability (*Pataky et al., 2014*).

Landmarks and other low-dimensionality representations of shape—including harmonic coefficients from elliptic Fourier analysis (*Bonhomme et al., 2014*)—embody a second important limitation: a potentially over-simplified representation of shape. In the case of landmarks, a danger of over-simplification arises from the Nyquist theorem: under-sampling a continuous process (including the continuous spatial surface of an object) can lead to aliasing, whereby the under-sampled measurement can misrepresent the true characteristics of the underlying object (*Nyquist, 1928*), and can even reverse statistical interpretations through mechanisms such as regional conflation (*Pataky et al., 2008*). This latter problem of shape simplification can nevertheless be solved by the use of semi-landmarks (*Bookstein, 1997*; *Adams, Rohlf & Slice, 2004*) which, as argued above, can be regarded as a specific approach to shape registration, implying that semi-landmark approaches could interface easily with the proposed technique.

An advantage of the proposed method is processing speed. The current, non-optimized analyses executed in under 2 s, with statistical inference itself requiring well under 100 ms (Table 3). We acknowledge that other data processing steps, including image segmentation and registration for example, can require substantial effort, so we caution readers that the reported execution speeds do not necessarily translate to reduced laboratory hours. The primary advantage in our view is instead the promotion of sensitivity analysis: since the entire data processing chain can be executed relatively rapidly, it would be possible to systematically adjust algorithm parameters, and even swap algorithms, in a sensitivity loop, to probe the robustness of particular results.

Another advantage of the proposed method is implicit morphological information. The proposed method yields results that are rich in morphological detail (Fig. 8) which, much like a highlighted photograph or x-ray image, can be readily interpreted at a glance. Since SPM operates directly on (registered) contours, without reducing the object-of-hypothesis-testing to a single abstract metric (like Procrustes ANOVA), or to a small handful of abstract metrics (like elliptical Fourier analysis), SPM results embody morphological meaning insofar as contours themselves embody morphological meaning. While individual contour points do not necessarily embody meaning, one could argue that the set of all contour points collectively embodies substantial morphological meaning. This perspective is analogous to a pixel-and-image argument. The color of a single pixel is largely irrelevant to the overall interpretation and meaning of an image. Similarly, the test statistic value at a single contour point is itself largely irrelevant to the overall morphological interpretation of SPM results;

morphological meaning is instead encapsulated implicitly in the overall excursion set, where 'excursion set' means the set of supra-threshold contour points, like those in Fig. 8. Regardless of the quality of morphological meaning, SPM results must be viewed as just one set of results, which may or may not embody useful morphological information, and which should be considered along with other, more explicit morphological methods like Procrustes ANOVA and elliptical Fourier analysis.

Considering last specific results from this paper, a particularly unintuitive set of results was observed for the Device8 dataset, for which UV analysis yielded the smallest $p$ value (0.022), and for which no other method yielded significance ($p > 0.2$) (Table 2). This result was likely caused by widespread but relatively small-magnitude mean-shape differences (Fig. 8C); since the deformation is widespread it would be detected by a general deformation metric like Procrustes distance, but since the deformation magnitude is relatively small it would not be detected by local contour-point methods like SPM. The interpretation is emphasized in the Flatfish dataset, where general deformations were similarly broadly distributed across the contour, but maximal local deformations were greater (Fig. 8E), which yielded significance in all methods (Table 2). Nevertheless, this interpretation appears to be inconsistent with the Horseshoe dataset, which exhibited both large and widely distributed deformation (Fig. 8H), but which also failed to yield significant UV results (Table 2). Nevertheless, this apparent consistency may be resolved by considering the large variability in the Horseshoe dataset, particularly at the selected landmarks (Fig. 2H). To more completely resolve such apparent inconsistencies, and more generally to understand the nature of landmark- vs. contour-based methods, it would be necessary to consider individual contour points, their deformations, and their covariances.

## Generalization to 3D analysis

While this paper was limited to 2D analysis, it should be noted that the proposed analysis framework (Fig. 1D) can be readily extendable to the morphological analysis of 3D surfaces. Similar to the unwrapping of 2D contours onto a 1D domain $u$ (Fig. 6), 3D surfaces can be unwrapped onto a 2D domain $uv$ Fig. 10, and methods like SPM (*Friston et al., 2007*) can be used to conduct domain-level hypothesis testing regarding these unwrapped data. This domain-wide testing is possible due to the underlying model of domain-level variance, which SPM models as smooth, Gaussian random fields, and which can be extended to arbitrarily high-dimensional domains with arbitrary geometry (*Adler & Taylor, 2007*). For the current paper involving 2D shapes, the (flattened) domain is one-dimensional, and the dependent variable is a two-component position vector; that is, a two-component position is defined at all locations $u$ along the contour. Similarly, for 3D surfaces, the (flattened) domain is two-dimensional and the dependent variable is a three-component position vector, where position is defined at all locations $uv$ across the surface. A variety of computational tools exist for 3D geometry flattening (e.g., *Dale, Fischl & Sereno, 1999*; *Sawhney & Crane, 2017*), so 3D implementations of the proposed method could presumably proceed in a fully automated manner.

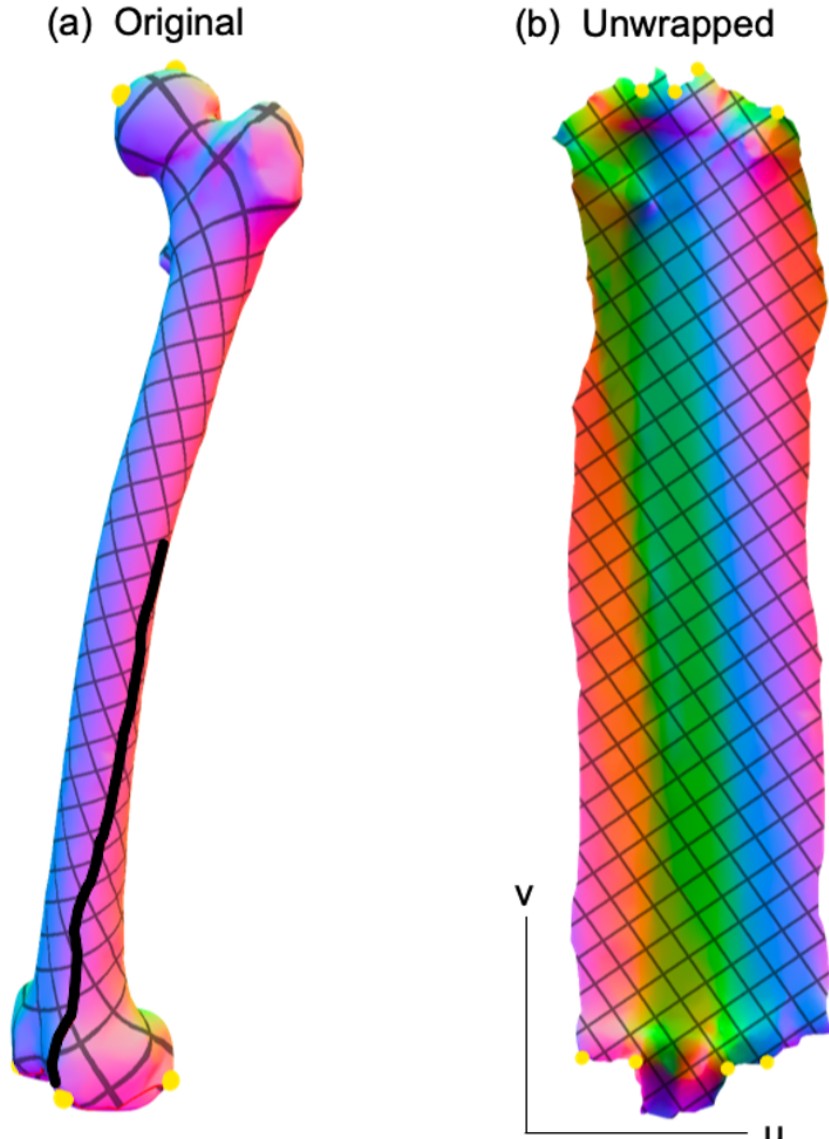

**Figure 10** **Example 3D surface unwrapping.** (A) Original 3D geometry. (B) Unwrapped geometry; this is a 2D parametric (UV) representation of the original geometry. Colors represent changes in surface normal direction. The thick black line in (A) represents a seam along which the 3D geometry is cut so that it can be flattened into a 2D shape. Unwrapping was performed here using boundary first flattening (*Sawhney & Crane, 2017*).

## Limitations

The proposed mass-multivariate framework (Fig. 1D) has a number of limitations. The most severe of these is sensitivity to algorithmic specifics. For example, simply by randomly changing the order of the points, it is possible to yield qualitatively different results (Fig. 11). Systematic, random variations of point ordering would be necessary for assessment of the results' sensitivity, but in our view this would be insufficient because ultimate results may also be sensitive to other particulars including, for example, specific parameter values used

in contour parameterization, registration, and correspondence algorithms. In other words, one should regard the results as potentially sensitive to all data processing steps, and not just to point ordering. The current paragraph considers just one example (point ordering) as a potential source of sensitivity concern. In (Fig. 11), the qualitative change in results can be attributed to a minor shift in point correspondence (Figs. 11A–11B), which created a small shift in pointwise covariance, but a shift that was large enough to alter the hypothesis rejection decision at $\alpha = 0.05$. That is, point-specific covariance is direction dependent, so small changes in point-deformation direction can yield qualitative changes in test statistics (Pataky et al., 2014). Nevertheless, we observed this type of sensitivity to random point ordering only occasionally, with most randomizations resulting in qualitatively similar results. Also, in most cases we noticed that probability results, while variable, were generally stable. The problem only emerged qualitatively when that variability spanned $\alpha=0.05$, as depicted in Fig. 11). This problem of probability value variability (Halsey et al., 2015) partially reflects a weakness of classical hypothesis testing, which has a binary interpretation of continuous probability. We acknowledge that we did not systematically conduct sensitivity testing, and also that each stage of processing involves a variety of components or parameters that could be subjected to sensitivity analysis. Comprehensive consideration of this sensitivity would require a large research effort, so we leave this for future work.

The datasets and analyses presented in this paper also have limitations. We analyzed shapes from just one database (Carlier et al., 2016) and, for each dataset, we selected only ten shapes for analysis, and only conducted two-sample tests. While we do not expect analysis of datasets from other databases to appreciably affect this paper's messages, we acknowledge that analyses of relatively small samples, and just one simple experimental design, fully exposes neither the advantages nor disadvantages of the proposed analysis framework. We selected just ten shapes for each dataset primarily to emphasize that the proposed parametric procedure is sufficiently sensitive to detect morphological effects for small sample sizes. The specific ten shapes were selected in an *ad hoc* manner to emphasize particular concepts including, for example: interpretation agreement between the proposed and landmark methods' results, and the opposite: interpretation disagreement. Since these datasets were selected in an *ad hoc* manner, from a single database, and with only two-sample analyses, the reader is left to judge the relevance of these results to other datasets and experimental designs.

## CONCLUSIONS

This paper demonstrates that parametric hypothesis testing can be conducted at the whole-contour level with suitably high statistical power for the analysis of even relatively small samples of 2D shapes ($N = 10$). We describe a general framework for automated, landmark-free hypothesis testing of 2D contour shapes, but this paper implements just one realization of that framework. The main advantages of the proposed framework are that results are calculated quickly ($<2$ s in this paper), and yield morphologically rich results in an easy-to-interpret manner. Since innumerable realizations of the proposed framework

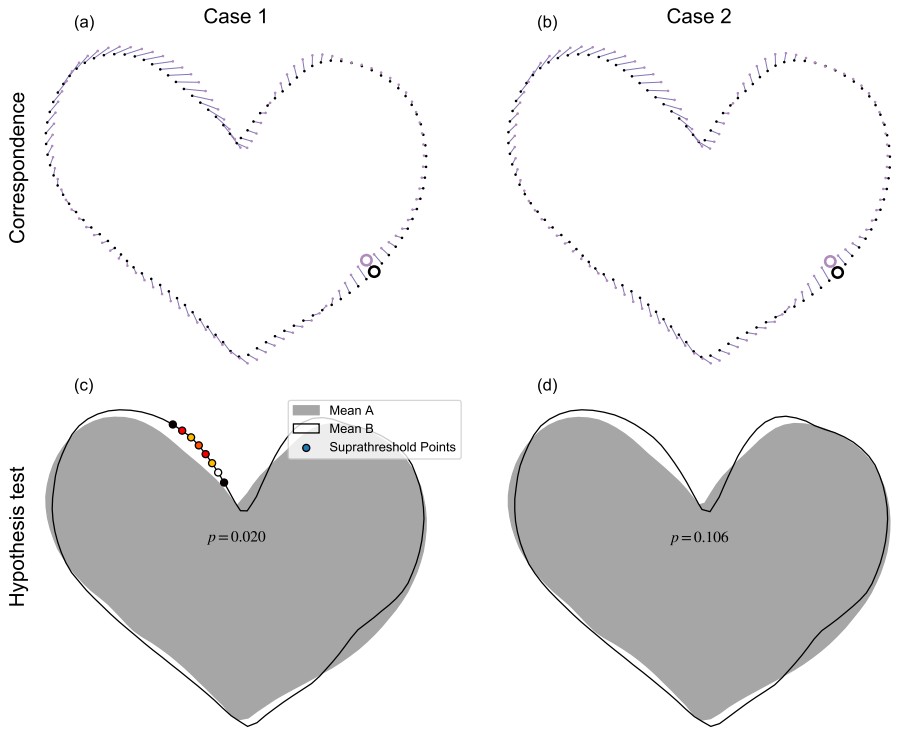

**Figure 11** **Example processing sensitivity.** Case 1 (A–B) depicts the result reported in Fig. 8G. Case 2 (C–D) depicts the results after point re-shuffling (i.e., a new random points order, see Fig. 5A), then re-application of the processing chain depicted in Fig. 1D. Note: results for Case 1 were qualitatively replicated for most random re-shufflings, but approximately 1 in 20 re-shufflings yielded qualitatively different results, like those depicted for Case 2.

are possible through algorithm and parameter substitution at each stage in the proposed data processing chain, sensitivity analysis may generally be required for robust statistical conclusions.

### Funding
This work was supported by Kiban B Grant 17H02151 from the Japan Society for the Promotion of Science. There was no additional external funding received for this study. The funders had no role in study design, data collection and analysis, decision to publish, or preparation of the manuscript.

### Grant Disclosures
The following grant information was disclosed by the authors:
Japan Society for the Promotion of Science: 17H02151.

### Competing Interests
Philip G. Cox was an Academic Editor for PeerJ.

## Author Contributions

- Todd C. Pataky and Philip G. Cox conceived and designed the experiments, performed the experiments, analyzed the data, performed the computation work, prepared figures and/or tables, authored or reviewed drafts of the paper, and approved the final draft.
- Masahide Yagi and Noriaki Ichihashi conceived and designed the experiments, analyzed the data, prepared figures and/or tables, authored or reviewed drafts of the paper, and approved the final draft.

## Data Availability

All data and code, including scripts that replicate all main results from the article, are available at GitHub: https://github.com/0todd0000/lmfree2d.

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
