# Peer review of "Landmark-free, parametric hypothesis tests regarding two-dimensional contour shapes using coherent point drift registration and statistical parametric mapping"

_PeerJ Computer Science, doi:10.7717/peerj-cs.542_

## Round 0.1 · original submission · Major Revisions

Please conduct a major revision and address all the comments from the reviewers point by point.

·

Basic reporting

The manuscript is written in clear, unambiguous, and professional English. The overall structure of the manuscript meets PeerJ standards, although some minor deviations might have been missed by this reviewer. The figures chosen are relevant, high quality, and well labelled. However, there are a couple additions that could be made in order to increase the readers ability to interpret the results and various claims of methodological value made by the authors. While the introductory and background sections do provide important context for this research, some references are missing, and a variety of similar methods have been totally missed by the authors.

First, the authors should add the missing references to lines 70 and 93.

Second, the authors explicitly focus on the weaknesses of Procrustes based GM analyses and the strengths of their contour-based MV methods (with some statements of contour method weakness in the discussion that are largely brushed aside). This is nowhere near a fair and balanced comparison for readers who may be unfamiliar with both methods. Of course, as stated by the authors, it is not appropriate to review all the detailed literature of GM here in this research article. But, the authors should take at least a few sentences to describe the scenarios in which GM methods are valuable, including those where biological homology (either morphological or developmental) is critical for interpretation of results. These are situations for which randomly identified points on a contour cannot provide the information necessary to interpret results effectively. Some relevant examples are likely found in the already cited Zelditch, et al., 2012 textbook. Other arguments for the value of explicit homology for interpretations of biological shape space can be found in recent reviews of GM within Biology such as Hallgrimsson, et al, 2015 (https://doi.org/10.1016/bs.ctdb.2015.09.003). Explicit recent reports on the strength of Procrustes ANOVA and related analyses within the study of multivariate shape data have been written by Adams and Coyler, among others (e.g., https://doi.org/10.1038/hdy.2014.75). The authors should (at minimum) acknowledge and cite previous papers that describe the value of these methods and the contexts within which they are most useful.

Third, there is no introduction of other non-Procrustes Distance based contour morphometric comparative methods (e.g., elliptic fourier analysis described in Bonhomme, et al., 2014 (https://hal.inrae.fr/hal-02635896) and Claude, 2013, Hystrix 24(1):94-102) or automated landmark correspondence quantification methods that have been previously published (e.g., Peter Claes’s 3D facial surface automated methods (https://www.esat.kuleuven.be/psi/members/00041773). While the MV method described in this manuscript is interesting and potentially useful as an alternative method to sparse landmark GM, it is not the first method that has been suggested as an alternative. Related and relevant previously published methods should be acknowledged. A literature dive to identify the range of these previously published methods and research is strongly suggested so that the authors can provide a broader theoretical and methodological context for the currently proposed approach. This will also allow the authors to compare their 2D contour method (and proposed 3D extension) with other contour and surface-based alternatives to GM (such as the Claes et al., methods) within their discussion.

Fourth, the fact that homologous landmarks have been collected manually (as stated on line 50) does not mean that automated methods are impossible. The issue until recently has been that automated methods are simply bad at successfully identifying biological relevant homologous points across a broad range of morphology. However, there are recent pushes to use modern nonlinear registration and machine learning to improve the quality of automated homologous landmark identification. This recent work should be acknowledged (e.g., https://doi.org/10.1007/s11692-020-09508-8 and https://doi.org/10.1186/s12983-015-0127-8).

Finally, a raw dataset of contours and landmark points is not provided. It is true that the authors used contours from a previously published dataset which is (presumably) publicly available. However, in order to replicate their results, a complete list of the 10 (out of 20) chosen shapes for each shape category is required. It would also be necessary to report which of those 10 shapes were placed in group A and placed in group B for each category. The best way to allow for replication would be to include a raw dataset that contains only the contours they analyzed, the group identities of each contour, and the manually identified landmarks of each contour.

Experimental design

The research within this manuscript appears to fit within the scope of the journal, although it largely represents a combination of previously published methods into a novel analysis pipeline. The authors do a good job identifying the purpose of their new pipeline and their comparison of relevant statistical methods. It appears that the authors have completed the work at a high technical and ethical standard. In most cases, the authors have done a great job describing the methodological details. However, there are a few places where further detail will be helpful to readers.

Line 117 – Since there are 20 shapes within each category that were presumably produced using explicit randomization parameters, why not have two random groups of 10 shapes. In other words, for each category, why were two groups of 5 chosen and explicitly how were they chosen? Were they chosen randomly or were they subjectively chosen in an effort to produce two groups that would probably display a significant difference in shape?

Line 148 – It would be useful to indicate what data is being used as the input for the T2 statistic. Two dimensional coordinate data is implied later in the section, but the input data should be made explicit. Also, it is not immediately clear what is meant by T2max. Is this the maximum T2 value between group A and B across all landmarks (which the results imply) or is this the maximum difference between specimens as calculated with a summary value of all landmarks within a specimen?

Line 165 – Certainly, a dataset of unordered points is possible within an automated phenotyping framework. However, it seems unlikely that 2D contour points generated within a real world scientific framework would be unordered, because there is clearly an order in position along the contour (i.e., it is a fairly simple ~1.5 dimension object). Are there examples of common real-world research datasets where the contour points are usually (i.e., generally) unordered at the beginning of the statistical analysis?

Line 175 – In order to complete a valid Procrustes ANOVA of automatically identified contour points, it would be necessary for all contours to be represented by the same number of points. At what stage in the Contours UV analysis is the number of points within all shapes equalized? Is there an interpolation to increase the number of points in all shapes to match the contour with the largest number of points (as implied in this passage)? If not, then which points are chosen to be included in the Procrustes analysis and how?

Validity of the findings

The underlying data and results appear robust, statistically sound, and controlled. The results are presented plainly and clearly, which is appreciated. There are a couple minor issue in the results that can be quickly fixed.

Table 2 – Face UV Contours p-value of 0.052 is not actually lower than the 0.050 cutoff. This should be corrected in the table and associated text

Line 215 – Table number is missing

Moving into the Discussion, most statements are reasonable suppositions from the presented results. The author’s attempt to present the data at face value for the judgement of readers is somewhat successful. In addition to highlighting some strengths of their proposed pipeline, the authors do mention a few weaknesses. This is also appreciated. However, in some parts of the discussion, the strength of the proposed pipeline and the weakness of Procrustes ANOVA analysis are overexaggerated to the point of being misleading. These statements should be amended to more accurately reflect the general value of these methods.

Line 226-229 – It is shown that the parametric T2 method and the nonparametric method identify some of the same significant differences. This is true. However, this does not show that parametric hypothesis testing is valid for these types of datasets. The author’s statement that their use is “possible” implies that it is statistically valid and appropriate to use for these datasets. If the authors want to say that parametric approaches are statistically valid and appropriate, they need to show that the input data for the T2 test actually meets the assumptions of a T2 distribution. If not, then they need to be clear that this may not be a statistically valid approach even if it does identify some of the same significant differences in shape as Procrustes ANOVA.

Line 232 (point 1) – It is true that manual landmark placement requires subjective judgement on the placement of landmarks on a surface or contour. In this way, it is more subjective than the placement of landmarks using the automated method proposed here. However, the choice of parameters and procedures for the automated landmark placement algorithm are also subjective. In addition, the fact that a given version of the algorithm produces 100% consistent landmark placement (i.e., high repeatability) does not necessarily mean that the automatically identified landmarks provide accurate, useful, or interpretable results. The authors should acknowledge that this method (as described here) might be more useful for some datasets and research questions than others. In other cases, care should be taken by researchers to validate that this “objective” method quantifies shape in a useful and interpretable way.

Line 232 (point 2) – The results do not been show that this method generally provides “direct morphological meaning in hypothesis testing results.” Yes, the T2 method does produce a p-value for all points along a contour, because a test is completed for each contour point. This is true. However, because those points have been randomly placed (rather than being morphologically defined homologous points of biological meaning), the points have no inherent meaning. It is absolutely still necessary to produce a visualization of significant landmark location along the contour (as done in Figure 8) in order to begin interpreting the meaning of those significant landmark coordinate differences. Both the Procrustes ANOVA and the proposed method require additional steps of visualization and follow up tests to generate valuable interpretations of results. Furthermore, randomly identified points along a contour have no inherent biological, morphological, developmental, or other meaning while well-defined homologous (and “subjective” manually placed) landmarks or semi-landmarks often do.

Line 255 – The authors correctly state that manual landmark placement can be very time consuming. However, they also imply that this contour based method requires much less researcher time to complete. This is not true in this reviewer’s experience with landmarking, segmentation, and contour definition. The authors have completely ignored the amount of time that it takes to produce the contours upon which automated landmarks are placed in a real world research scenario. On a 2D image, manually identifying landmarks might take many minutes to identify carefully. However, identifying a sparse set of landmarks on an image is much less time consuming than drawing an accurate contour on a 2D image. This is a major reason why landmark based methods, rather than contour-based methods have been preferred in the past by researchers. Even if a semi-automated method is used to identify a contour (such as with edge detection), a researcher usually needs to open up every image in order to verify that the automated contours are accurate and often needs to make at least a few corrections to each contour. So, this reviewer agrees that the automated placement of points along an already defined contour using the described method is faster than manually identifying landmarks on an already defined contour. But, this will not translate into real world time savings for researchers in the way that the authors imply. The time and effort required to produce accurate contours in real world research situations needs to be acknowledged.

Line 277 & Figure 8 – The authors correctly point out that the proposed multivariate method fails to identify differences in group A and group B shape across many of the shape categories, including around the horseshoe, the belly of the fish, differences in the head of key, and the top of the bell. As mentioned, this is likely because the drawn mean shapes in Figure 8 do not represent the full scope of the shape variance within group A and group B of each category. It is highly likely that high contour shape variance explains 1) the failure of the multivariate method to identify these horseshoe differences AND 2) the failure of the UV method to identify significant shape differences for several of the shape categories. However, in order for the reader to easily judge this possibility, they need to have a visualization of the shape variance. To clarify this issue, the authors should include the outlines of group A mean, group B mean (perhaps as thick colored lines) and all individual shape contours (perhaps as thin dotted colored lines) within Figure 8’s panels. This will improve reader understanding of the dataset and improve the strength of the discussion about why each method failed to identify significant shape differences in some cases.

Line 295 – As mentioned in my comments about the introduction, previous methods of 3D automated landmarking and analysis like the Claes methods should be cited here.
Line 297-305 – Given the described sensitivity to algorithmic specifics, would the authors recommend that multiple separate MV tests be run with random point location seeds? This might allow researchers to identify whether or not their shape is at ~0.05 p-value and to determine the typical level of significance for the shape. Computationally, this should be possible given the small amount of time required for each run.

Additional comments

Generally speaking, this reviewer was glad to read this paper, found the quality of the results presentation to be high, and will genuinely consider the automated contour point identification method for future use.

·

Basic reporting

Line 79: It is absolutely incorrect to claim that geometric morphometric analyses are generally univariate, relying on analysis of PCs. There are innumerable examples that show this is not correct and as such this statement suggests that the authors do not know the GM literature. It is also not true that a univariate Procrustes ANOVA is generally used to test for shape differences. The criticism of Procrustes ANOVA and RRPP is also unnecessary for this paper. Validating a robust landmark-free method would be tremendously useful and entering into what appears to be an insufficiently well informed (or at least insufficiently supported) critique of the statistical hypothesis testing in GM is an irrelevant distraction for this paper.

A major issue is that it is not clear how one might use this form of morphometric qualification beyond the simple task of statistical comparison of mean shape and the visualization of differences in shape between groups. This is only the most basic of morphometric tasks. A key requirement for morphometric methods is the ability to extract measures of shape variation that can be quantitatively compared to covariates. For example, the regression of shape on size to quantify allometry is a basic task in most morphometric analyses. Similarly, regression on covariates that one wishes to control for or remove from an analysis is a basic requirement. Finally, much of GM focuses on analysis of covariance structure with methods such as PLS used to quantify covariation in shape between parts of an organism. There are existing landmark-free methods that enable comparison of means and visualization of mean differences. Most fail to be adopted by the morphometric community because they fail to allow for analyses of integration, covariance structure or analyses involving multiple covariates. The claim is made in the paper that the method is amenable to standard (including multivariate statistical) techniques such as regression. This is a critical point and needs to be developed further. How might one do a regression of shape on size, visualize the resulting variation and then remove the covariance of shape on size prior to further analyses? For this method to be viewed as promising by the morphometric community, the pathway to such analyses must be clear.

Experimental design

I have no concerns in this area.

Validity of the findings

The validation is robust as far as it goes, although see more general concerns as outlined above.

---

## Round 0.2 · Minor Revisions

Please see the attached comments. The reviewer still has some concerns. Please revise accordingly by providing the detailed point-by-point rebuttal.

·

Basic reporting

113-131: More details on group definition and shape choice are appreciated. However, as written, information in these three paragraphs is presented in a confusing order. Please reorder and edit to improve reader understanding. For example, at one point, it is claimed that shapes are assigned to groups in a pseudo-random manner. Presumably, this is after 10 shapes were chosen randomly for each object type (as described in the next paragraph). Then, in the last paragraph is described explicitly or is this something else?

Figure 8: Based on other tables, it appears that the parametric p-values are found in dark grey shapes and the nonparametric p-values are found in light grey shapes. However, this should probably be indicated explicitly within the Figure caption.

280: imply instead of apply?

Discussion: Perhaps subsection headings of the discussion would be useful. For example (~Line 374), to separate more general discussion of the SPM method from discussion of specific results presented in the paper?

433: ad hoc instead of add hoc?

Experimental design

201: The clarification of contour points is helpful. However, the statement that the shape with the maximum number of points is used as a template for shape registration is still potentially confusing within the context of a standard Procrustes ANOVA analysis. Within GM, Procrustes superimposition of shapes would be based on an equal number of 1:1 homologous points per shape. The number of points is one of several factors that will lead to different Procrustes distance estimations. So, if a standard Procrustes ANOVA is being conducted, the reader assumption is likely to be that Procrustes distances were estimated between shapes represented by equal numbers of points. The author response to previous comment 1.11 is comforting to the reviewer, but will not be seen by other readers. The statement that the shape with maximum points (even if this means maximum number of points before parameterization) will be off-putting to GM readers. Can this implication of shapes with different numbers of points within Procrustes ANOVA be clarified?

Validity of the findings

114-116 & other places: The clarification that one major goal is to show statistical power to identify differences using parametric methods is helpful here and elsewhere. As someone who applies existing methods more often than he develops statistical methods, this reviewer previously assumed that phrases like “suitability” and “possible” did imply empirical appropriateness of methods rather than being specifically related to statistical power. The explicit clarification of meaning that have been made throughout the paper are much appreciated.

However, in this reviewer’s experience/interpretations of various readings, the main concern about parametric testing methods for multivariate morphometric data from small datasets has been that the assumptions of parametric methods (including distribution assumptions) may not be met. This leads to a concern that statistically significant signals generated with a parametric test are invalid and may not reflect a “real” difference between groups under study. Of course, “real” means different things in different contexts. But, the idea of statistical power seems secondary to the idea of empirical validity to this reviewer (who is acknowledges a potential blindspot on this subject). That being stated, the explicit statements of word meaning and research goals added by the authors in the revised manuscript are certainly adequate to prevent reader confusion. This potential philosophical difference is no reason to delay publication and requires no practical changes to the manuscript.

Additional comments

The authors have made major revisions and welcome additions to the Introduction and Discussion session that address most previous reviewer comments. These changes will make this work more accessible and increase its impact among practicing morphometricians. The in-depth comments, clarifications, and illustrations of statistical concepts within the author response are helpful and much appreciated. A few potential issues remain, but they are minor in nature.

---

## Round 0.3 · accepted · Accept

I have no further concerns regarding the publication of this manuscript.

·

Basic reporting

no comment

Experimental design

no comment

Validity of the findings

no comment

Additional comments

I have no further concerns regarding the publication of this manuscript.

·

Basic reporting

No comment

Experimental design

No comment

Validity of the findings

No comment

Additional comments

This is an interesting addition to the literature. In this revision, the authors have done a much better job of contextualizing this new method and exploring its limitations (which are really just areas for further development) and advantages. One thing I would like to see in the future is validation by comparison to GM approaches (sparse and dense landmark) on real-world datasets that have been used for hypothesis testing. However, the work as presented here stands on its own and so should get out there for people to consider and digest.